# Characterization of Gene Expression Suppression by Bovine Coronavirus Non-Structural Protein 1

**DOI:** 10.3390/v17070978

**Published:** 2025-07-13

**Authors:** Takehiro Ohkami, Ichika Kitashin, Riko Kawashima, Aimi Yoshida, Taizo Saito, Yasuhiro Takashima, Wataru Kamitani, Keisuke Nakagawa

**Affiliations:** 1Laboratory of Veterinary Microbiology, Joint Department of Veterinary Medicine, Gifu University, Yanagido, Gifu 501-1193, Japan; 2Joint Graduate School of Veterinary Sciences, Gifu University, Yanagido, Gifu 501-1193, Japan; yoshida.aimi.i4@s.gifu-u.ac.jp (A.Y.);; 3Laboratory of Infectious Diseases, Joint Department of Veterinary Medicine, Gifu University, Yanagido, Gifu 501-1193, Japan; 4Education and Research Center for Food Animal Health, Gifu University, Yanagido, Gifu 501-1193, Japan; 5Laboratory of Veterinary Parasitology, Joint Department of Veterinary Medicine, Gifu University, Yanagido, Gifu 501-1193, Japan; 6Center for One Medicine Innovative Translational Research, Gifu University, Yanagido, Gifu 501-1193, Japan; 7Department of Infectious Diseases and Host Defense, Graduate School of Medicine, Gunma University, Gunma 371-8511, Japan

**Keywords:** bovine coronavirus, non-structural protein 1, shutoff function, ribosome

## Abstract

Coronavirus non-structural protein 1 (nsp1) is a pathogenic determinant of *Betacoronaviruses*. Previous studies demonstrated that the nsp1 of various coronaviruses induces host shutoff through a variety of mechanisms; however, there is little information on the function of bovine coronavirus (BCoV) nsp1. We aimed to characterize the host gene expression suppression function of BCoV nsp1. We first confirmed that the expression of BCoV nsp1 in MAC-T cells, a bovine mammary epithelial cell line, suppressed host and reporter gene expression. Subsequently, lysine and phenylalanine at amino acid positions 232 and 233, respectively, were identified as key residues required for this suppressive effect. Expression levels of housekeeping genes are comparable in cells expressing wild-type BCoV nsp1 and a mutant with alanine substitutions at positions 232 and 233 (BCoV nsp1-KF). Wild-type BCoV nsp1 localized to both the cytoplasm and nucleus; however, BCoV nsp1-KF exhibited prominent nuclear accumulation with dot-like structures. Using confocal microscopy and co-sedimentation analysis, we identified an association between wild-type BCoV nsp1, but not BCoV nsp1-KF, and ribosomes, suggesting that ribosome binding is required for BCoV nsp1-mediated suppression of host gene expression. This is the first study of the characterization of host gene expression suppression by BCoV nsp1.

## 1. Introduction

Bovine coronavirus (BCoV) is an etiological agent of respiratory disease and winter dysentery in cattle. The virus has a high affinity for mucosal epithelial cells of the bovine intestinal/respiratory tracts and occasionally causes mixed infections with other pathogens, including bovine rotavirus, cryptosporidium, and toxigenic *Escherichia coli*, leading to a fatal outcome [1,2]. BCoV is widespread in cattle of all ages, resulting in economic loss to the beef and dairy industry worldwide [3,4]. To control BCoV-mediated diseases, comprehensive studies on its epidemiology and vaccine development have been conducted worldwide [5,6,7,8,9,10,11,12,13,14,15,16,17,18]. However, there is little information on viral factors involved in the pathogenicity of BCoV.

BCoV belongs to the species *Betacoronavirus* 1 (subgenus Embecovirus) of the *Betacoronavirus* genus in the subfamily Orthocoronavirinae and family Coronaviridae. BCoV is an enveloped RNA virus with a large (~30 kb), capped and polyadenylated, positive-sense single-stranded RNA genome [19,20]. The 5′ two-thirds of the genome (ORF1ab) encodes gene 1 proteins, whereas the 3′ one-third encodes structure and accessory proteins. BCoV particles possess five major structural proteins: the nucleocapsid (N) protein, the membrane (M) protein, the small membrane/envelope (E) protein, hemagglutinin-esterase (HE), and the spike (S) protein. The latter consists of an S1 subunit that contains the receptor-binding domain and an S2 subunit that mediates viral membrane fusion. HE acts as a receptor-destroying enzyme (esterase) to reverse hemagglutination. The N protein is located inside the viral envelope and is associated with the viral RNA genome. The M protein spans the viral envelope, while the S protein and HE project from the envelope. Upon entry into host cells, the coronavirus genomic RNA is released into the cytoplasm. It translates two large, partially overlapping precursor polyproteins from ORF1ab. Two virally encoded proteinases proteolytically process these precursor polyproteins to generate 16 mature nonstructural proteins (nsp1–nsp16) in *Betacoronaviruses* [21]. All of these nsps, except for nsp1 [22] and nsp2 [23], are considered to be essential for viral RNA synthesis [24].

Among the coronavirus proteins, nsp1 is known as a multifunctional protein [25,26,27,28,29,30,31] and a virulence factor in several coronaviruses: transmissible gastroenteritis virus (TGEV), mouse hepatitis virus (MHV), mouse-adapted SARS-CoV, and SARS-CoV-2 [32,33,34,35,36]. In particular, the nsp1 of SARS-CoV or SARS-CoV-2 has been the most extensively studied for its role in suppressing host gene expression [37,38,39]. Kamitani et al. reported that SARS-CoV nsp1 stably binds to the 40S ribosomal subunit and inactivates the translational function of the 40S ribosomal subunit, leading to inhibition of protein synthesis [38]. In addition, SARS-CoV nsp1 also induces the degradation of endogenous host mRNAs in expressed cells and in SARS-CoV-infected cells [28,40], further contributing to inhibition of host gene expression, including type I interferon (IFN) expression. In other coronaviruses, including MERS-CoV, PEDV, TGEV, and FIPV, nsp1 has also been shown to inhibit host and reporter gene expression [29,32,41,42]. These studies strongly suggest that nsp1 is a pathogenic determinant in various coronavirus species. However, to the best of our knowledge, there is no report on the function of BCoV nsp1 in host gene expression suppression.

In the present study, we investigated whether BCoV nsp1 has a function of gene expression suppression in bovine cultured cells. The expression of wild-type BCoV nsp1 (BCoV nsp1-wt) induced strong inhibition of exogenous and endogenous gene expression. Also, we identified that lysine and phenylalanine at positions 232 and 233 of BCoV nsp1 are important for the suppression of gene expression. We named the loss-of-function mutant with alanine substitutions at positions 232 and 233 of BCoV nsp1 BCoV nsp1-KF. Wild-type BCoV nsp1-wt and BCoV nsp1-KF localized to both the cytoplasm and nucleus; however, BCoV nsp1-KF exhibited prominent nuclear accumulation with a dot-like structure. Finally, we demonstrated that the ability of BCoV nsp1-wt to bind to ribosomes is associated with the inhibitory effect on host gene expression. This is the first study of the characterization of host gene expression suppression by BCoV nsp1.

## 2. Materials and Methods

### 2.1. Cells

MDBK cells (a cell line derived from the kidneys of adult cows, ATCC number CCL-22), MAC-T cells (a cell line derived from bovine mammary epithelial cells, which was kindly provided by Sanggun Roh at Tohoku University) [43], and HEK-293 cells (a cell line derived from human embryonic kidneys, ATCC number CRL-1573) were maintained in Dulbecco’s modified eagle medium (Fujifilm Wako, Osaka, Japan) with 10% fetal calf serum (Gibco, Life Technologies Corporation, Carlsbad, CA, USA) and 1% penicillin–streptomycin (Fujifilm Wako). The cells were maintained at 37 °C in a CO_2_ incubator.

### 2.2. Plasmids and Plasmid Construction

A Renilla luciferase-expressing plasmid, pRL-TK (Promega, Madison, WI, USA), and an eGFP-expressing plasmid, pCX-eGFP (kindly provided by Yasuhiro Takashima at Gifu University), were used in this study. The pCAGGS-based expression plasmids, pCAGGS-chloramphenicol acetyltransferase (CAT)-FLAG, -SARS-CoV-2 nsp1-wt-FLAG, and -SARS-CoV-2 nsp1-KH-FLAG, all of which contain a C-terminal FLAG tag, were described previously [44]. The plasmids pCAGGS-BCoV nsp1-wt-FLAG, pCAGGS-BCoV nsp1-KG-FLAG, pCAGGS-BCoV nsp1-KF-FLAG, and pCAGGS-BCoV nsp1-KK-FLAG were generated by conventional cloning techniques. pCAGGS-BCoV nsp1-KG-FLAG carries Lys-to-Ala and Gly-to-Ala substitutions at amino acid positions 230 and 231, respectively; pCAGGS-BCoV nsp1-KF-FLAG carries Lys-to-Ala and Phe-to-Ala substitutions at positions 232 and 233, respectively; and pCAGGS-BCoV nsp1-KK-FLAG carries Lys-to-Ala substitutions at positions 235 and 236. Briefly, EcoRI-digested pCAGGS-SARS-CoV-2 nsp1-wt-FLAG was used as a vector. A PCR product encoding BCoV nsp1-wt as an insert was generated using KOD Fx neo (TOYOBO, Osaka, Japan) with cDNA from the BCoV GF2020 strain and primers shown in Appendix A. The PCR products were inserted into the vector with an In-Fusion HD Cloning Kit (TaKaRa Bio Inc., Kusatsu, Japan). Also, to generate PCR products encoding BCoV nsp1 carrying specific mutations, we performed recombination PCR using the primers shown in Appendix A. The 2nd PCR products were inserted into the vector by the same methods. Detailed information on the construction of these plasmids is available from the authors on request.

### 2.3. Plasmid Transfection, Reporter Assay, and Western Blot Analysis

Subconfluent HEK-293 cells in a 12-well plate were co-transfected in triplicate with pRL-TK (Promega) and pCAGGS-based expression plasmids (1 μg total) using the TransIT-293 reagent (Mirus Bio, Madison, WI, USA) or Lipofectamine LTX Reagent (Invitrogen, Carlsbad, CA, USA). Subconfluent MAC-T cells in a 12-well plate were co-transfected in triplicate with pRL-TK and pCAGGS-based expression plasmids (0.5 μg total) using Lipofectamine LTX Reagent (Invitrogen). At 24 h post-transfection, cell lysates were prepared and subjected to Renilla luciferase (rLuc) reporter activity assays (Promega). For confirmation of protein expression by Western blot analysis, cell extracts were prepared in sodium dodecyl sulfate–polyacrylamide gel electrophoresis (SDS-PAGE) sample buffer. An anti-FLAG antibody (Fujifilm Wako) and an anti-tubulin antibody (Santa Cruz Biotechnology, Dallas, TX, USA) were used as primary antibodies. Goat anti-mouse immunoglobulin G–horseradish peroxidase (BIO-RAD, Hercules, CA, USA) was used as the secondary antibody.

### 2.4. SUnSET Assay and Viability Measurement

MAC-T cells were transfected with each pCAGGS-based expression plasmid using Lipofectamine LTX Reagent (Thermo Fisher Scientific, Waltham, MA, USA). The cells were incubated at 37 °C in complete media. At 24 h post-transfection, the media were replaced with media supplemented with 0.5 μg/mL of puromycin, and incubation was continued for 30 min at 37 °C. The cells were harvested in phosphate-buffered saline (PBS), pelleted via centrifugation, and lysed in SDS sample buffer. The lysates were analyzed via Western blot analysis using anti-puromycin, clone 12D10 (EMD Millipore Corporation, Darmstadt, Germany). Goat anti-mouse immunoglobulin G–horseradish peroxidase (BIO-RAD) was used as a secondary antibody. Moreover, the cell lysates were subjected to Western blot analysis using an anti-FLAG antibody and an anti-tubulin antibody as described above.

Cell viability was evaluated by measuring the level of adenosine triphosphate in the transfected MAC-T cells. At 24 h post-transfection, the cultured medium was replaced with D-MEM and incubated for 30 min at 37 °C in a CO_2_ incubator. The same volume of CellTiter-Glo (Promega) was mixed, and luminescence was measured after 10 min using a GloMax Discover Microplate Reader (Promega).

### 2.5. Immunofluorescence Staining and Confocal Microscopy Analysis

MAC-T cells, grown on an EZWIEW Glass Bottom Culture Plate LB (IWAKI, Shizuoka, Japan), were transfected with each pCAGGS-based expression plasmid using Lipofectamine LTX Reagent. At 24 h post-transfection, the cells were fixed with 4% paraformaldehyde in PBS for 10 min, permeabilized in 100% methanol for 1 min, and immunostained with an anti-FLAG antibody (Fujifilm Wako) and/or an anti-S6 antibody (Cell Signaling, Danvers, MA, USA). An Alexa Fluor 488- or 594-conjugated secondary antibody (Thermo Fisher Scientific) was used as the secondary antibody. Nuclei were stained with a Cellstain Hoechst 33342 solution (DOJINDO, Kumamoto, Japan). The cells were examined under a KEYENCE BZ-X800 series microscope (KEYENCE, Osaka, Japan) or a ZEN3 LSM900 confocal microscope (Carl Zeiss, Oberkochen, Germany).

### 2.6. Total RNA Extraction and qRT-PCR

MAC-T cells were transfected with each pCAGGS-based expression plasmid using Lipofectamine LTX Reagent. At 24 and 36 h post-transfection, total cellular RNAs were extracted from plasmid-transfected cells using TRIzol LS reagent (Invitrogen) and Direct-zol RNA MiniPrep (Zymo Research Corporation, Irvine, CA, USA), according to the instruction manuals. cDNAs were synthesized using SuperScript III reverse transcriptase and random primers (Invitrogen). qPCR was performed using Applied Biosystems StepOnePlus (Thermo Fisher Scientific) and PowerUp™ SYBR™ Green Master Mix (Thermo Fisher Scientific), following the instruction manuals. The purity of the amplified PCR products was confirmed by the dissociation melting curves obtained after each reaction. The following previously designed primers were used [45]: 5′-ACACCCTCAAGATTGTCAGCAA-3′ (forward) and 5′-GCATCGTGGAGGGACTTATGA-3′ (reverse) for bovine *GAPDH* mRNA, 5′-AACGACCAGTCAACAGGCGA-3′ (forward) and 5′-CTGATGAAAAGGACCCCTCG-3′ (reverse) for *HPRT1* mRNA, and 5′-CCTGCGGCTTAATTTGACTC-3′ (forward) and 5′-GTGCATGGCCGTTCTTAGTT-3′ (reverse) for 18S rRNA. To obtain copy number data of each mRNA in the cDNA by qRT-PCR, we generated a standard curve by using each housekeeping gene PCR product amplified with the primers shown in Appendix A. Based on the amounts and lengths of the PCR products, we calculated their copy numbers. All of the assays were performed in triplicate. The results are expressed as means ± standard deviations.

### 2.7. Co-Sedimentation Analysis

Lysates from MAC-T cells expressing CAT, BCoV nsp1-wt, or BCoV nsp1-KF were prepared in a lysis buffer containing 50 mM Tris-HCl (pH 7.5), 5 mM MgCl_2_, 100 mM KCl, 1% (*v*/*v*) Triton X-100, 2 mM dithiothreitol (DTT), 100 mg/L cycloheximide, and 0.5 mg/L heparin. The lysates were applied onto a 10% to 40% continuous sucrose gradient prepared in the same buffer and centrifuged at 38,000 rpm in a Himac P40ST rotor (Eppendorf Himac Technologies Company Limited., Ibaraki, Japan) at 4 °C for 3 h. After fractionation, the proteins in each fraction were precipitated with trichloroacetic acid/acetone and detected by Western blotting with an anti-FLAG antibody. Total RNAs were also extracted from the fractions, and the rRNAs were visualized by staining with ethidium bromide.

### 2.8. Statistical Analysis

One-way analysis of variance with Tukey’s multiple-comparison test was conducted to determine statistical significance. A *p* value of <0.01 was significant.

## 3. Results

### 3.1. Assessment of the Appropriateness of MAC-T Cells in Transfection Experiments

Previously, MAC-T cells demonstrated higher transfection efficiency and moderately higher cytotoxicity compared with MDBK cells, which are derived from adult bovine kidneys [46]. In the current study, we assessed the appropriateness of MAC-T cells for transfection experiments in terms of transfection efficiency and cytotoxicity (Figure 1). Consistent with previous reports, we confirmed that MAC-T cells have a much higher level of transfection efficiency than MDBK cells (Figure 1A). Next, we assessed cytotoxicity by transfecting with 400 ng of GFP-expressing plasmid (Figure 1B). At 24 h post-transfection, there was no significant difference in cell viability between mock-transfected MAC-T cells and GFP-expressing plasmid-transfected MAC-T cells (Figure 1C), suggesting that the cytotoxicity under the experimental conditions would not affect the experimental results. Based on these data, almost all transfection experiments in this study were performed under similar transfection conditions: analysis at 24 h post-transfection with about 400 ng of one or more plasmids in MAC-T cells.

### 3.2. Suppression of Plasmid-Driven Gene Expression by BCoV nsp1 and Subsequent Identification of Important Amino Acids of BCoV nsp1 for Its Gene Expression Suppression Function

Since the nsp1s of many *Betacoronaviruses* suppress gene expression, it was easily predicted that BCoV nsp1 would also possess a similar function. Additionally, it was reported that alanine substitutions of charged amino acid residues (Lys and His at positions 164 and 165 of SARS-CoV-2 nsp1; Lys at position 181 of MERS-CoV nsp1) impaired their gene expression suppression function [30,39]. Multiple alignment of the amino acid sequences of *Betacoronavirus* nsp1s (BCoV nsp1, SARS-CoV-2 nsp1, MERS-CoV, and mouse hepatitis virus) showed that Lys and Gly at positions 230 and 231, Lys and Phe at positions 232 and 233, or Lys and Lys at positions 232 and 233 may correspond to Lys and His at positions 164 and 165 of SARS-CoV-2 nsp1 (Figure 2A). Based on the above, we hypothesized that BCoV nsp1 has a host shutoff function and that alanine substitutions of charged amino acid residues near the C-terminal region of BCoV nsp1 would also disrupt its gene suppression function.

First, to determine whether BCoV nsp1 suppresses plasmid-driven gene expression, we co-transfected HEK-293 cells or bovine mammary epithelial MAC-T cells with the plasmid pRL-TK, which expresses the thymidine kinase promoter-driven rLuc gene, as well as one of the following plasmids: pCAGGS carrying the gene for CAT, BCoV nsp1-wt, SARS-CoV-2 nsp1-wt, or SARS-CoV-2 nsp1-KH. Notably, SARS-CoV-2 nsp1-KH does not suppress host gene expression [39]. All proteins contained a C-terminal FLAG epitope tag. The expression of both SARS-CoV-2 nsp1-wt and BCoV nsp1-wt efficiently suppressed rLuc activity in 293 and MAC-T cells; however, CAT and SARS-CoV-2 nsp1-KH did not (Figure 2B,D). This suggests that BCoV nsp1-wt inhibits plasmid-driven gene expression. The low accumulation level of BCoV nsp1-wt suggests that it may inhibit its own expression.

We also generated non-tagged BCoV nsp1-wt- and N-terminal FLAG-tagged BCoV nsp1-wt-expressing plasmids and evaluated their shutoff function for plasmid-driven gene expression (Appendix A). As a result, both non-tagged BCoV nsp1-wt and the N-terminal FLAG-tagged BCoV nsp1-wt suppressed rLuc activity in MAC-T cells, similar to C-terminal FLAG-tagged BCoV nsp1-wt. However, we could not obtain an image of the band for the N-terminal FLAG-tagged BCoV nsp1-wt. We suspect that the N-terminal FLAG-tagged BCoV nsp1-wt might exert a strong shutoff function, leading to self-inhibition of its own expression from the plasmid. Due to the difficulty in confirming protein expression from the N-terminal FLAG-tagged BCoV nsp1-wt-expressing plasmid, we decided to use the C-terminal FLAG-tagged BCoV nsp1-wt-expressing plasmid for the characterization of gene expression suppression.

Next, we aimed to identify the important amino acid regions of BCoV nsp1 that are required for inhibiting gene expression. We constructed pCAGGS-based BCoV nsp1 mutant expression plasmids, pCAGGS-BCoV nsp1-KG-FLAG carrying Lys-to-Ala and Phe-to-Ala substitutions at amino acid positions 230 and 231, respectively, pCAGGS-BCoV nsp1-KF-FLAG carrying Lys-to-Ala and Phe-to-Ala substitutions at amino acid positions 232 and 233, respectively, and pCAGGS-BCoV nsp1-KK-FLAG carrying Lys-to-Ala and Lys-to-Ala substitutions at amino acid positions 235 and 236, respectively, and evaluated their ability to suppress plasmid-driven gene expression. In HEK-293 cells (Figure 2B,C), none of the BCoV nsp1 mutants inhibited plasmid-driven gene expression. We confirmed the expression of each BCoV nsp1 mutant in the transfected cells by Western blot analysis. On the other hand, we found that BCoV nsp1-KG and -KK showed suppression of plasmid-driven gene expression in MAC-T cells (Figure 2D,E). Importantly, only BCoV nsp1-KF did not exert the function of the plasmid-driven gene expression in MAC-T cells. Taken together, our data demonstrated that Lys-to-Ala and Phe-to-Ala amino acid substitutions at positions 232 and 233 of BCoV nsp1 disrupt the function of plasmid-driven gene expression suppression in MAC-T cells derived from a susceptible host.

### 3.3. Expression of BCoV nsp1, but Not BCoV nsp1-KF, Suppresses Host Gene Expression in MAC-T Cells

To investigate the function of BCoV nsp1 in host gene expression suppression, but not plasmid-driven gene expression suppression, we performed a SUnSET assay [47] in MAC-T cells expressing BCoV nsp1-wt and BCoV nsp1-KF (Figure 3). Consistent with a previous report [47], treatment with CHX suppressed host protein synthesis (Figure 3A–C). BCoV nsp1-wt expression in MAC-T cells suppressed host protein synthesis. In contrast, we found that BCoV nsp1-KF expression did not inhibit host protein synthesis. We confirmed the expression of BCoV nsp1-wt and BCoV nsp1-KF by Western blot analysis with an anti-FLAG antibody. Furthermore, there were no differences in cell viability among MAC-T cells expressing BCoV nsp1-wt and BCoV nsp1-KF (Figure 3D). Taken together, these data demonstrate that BCoV nsp1 has the function of host gene expression suppression and that the Lys and Phe at positions 232 and 233 of BCoV nsp1 are required for the function in MAC-T cells.

### 3.4. Expression Levels of Housekeeping Genes in MAC-T Cells Expressing BCoV nsp1-wt and BCoV nsp1-KF

It is reported that SARS-CoV nsp1, MERS-CoV nsp1, and SARS-CoV-2 nsp1 induce the degradation of endogenous host mRNAs in expressed cells [29,38,39]. We investigated whether the expression of BCoV nsp1 also induced a reduction in housekeeping gene mRNAs (bovine *GAPDH* mRNA, bovine *HPRT1* mRNA, and r18S ribosomal RNA) in MAC-T cells using qRT-PCR (Figure 4A,B). We found no significant differences in housekeeping genes between MAC-T cells expressing BCoV nsp1-wt and BCoV nsp1-KF at 24 and 36 h post-transfection. This suggests that a reduction in host mRNAs does not contribute to the host gene expression suppression by BCoV nsp1 under the experimental condition.

### 3.5. Subcellular Localization of BCoV nsp1-wt and BCoV nsp1-KF in MAC-T Cells and Colocalization of BCoV nsp1 with Ribosome

We investigated the subcellular localization of BCoV nsp1-wt and BCoV nsp1-KF in MAC-T cells. In the orthogonal-view images (Figure 5A), while the majority of BCoV nsp1-wt was localized in the cytoplasm, a small amount was also detected in the nucleus. On the other hand, clear dots of BCoV nsp1-KF were observed in the nucleus (Figure 5B), although it was still found in the cytoplasm.

Furthermore, we investigated whether BCoV nsp1-wt was colocalized with ribosomes using a confocal microscope (Figure 6 and Appendix A), since SARS-CoV-2 nsp1 binds to the 40S ribosome to exert its biological functions [39]. We found that BCoV nsp1-wt, but not BCoV nsp1-KF, was colocalized with ribosomes. Thus, the subcellular localization of BCoV nsp1 was changed by the Lys-to-Ala and Phe-to-Ala amino acid substitutions at positions 232 and 233, probably due to a change in the binding property of BCoV nsp1 with the ribosome.

### 3.6. Co-Sedimentation of BCoV nsp1-wt, Not BCoV nsp1-KF, with the 40S Ribosomal Subunit

Next, we examined the association of BCoV nsp1 with 40S ribosomal subunits by sucrose gradient sedimentation analysis of extracts from MAC-T cells expressing BCoV nsp1-wt. Lysates from MAC-T cells expressing CAT served as a negative control. Most of the CAT was detected near the top of the gradient and did not co-sediment with the 40S subunit, which was determined by detecting 18S rRNA, a component of the 40S ribosomal subunit (Figure 7A,B). In contrast, BCoV nsp1-wt co-sedimented with the 40S peak (Figure 7C,D). Of note, BCoV nsp1-KF mirrored the profile observed for CAT (Figure 7E,F). Taken together, our results strongly suggest that the ability of BCoV nsp1 to bind to ribosomes is important for the exertion of host gene expression suppression.

## 4. Discussion

In several *Betacoronaviruses* and *Alphacoronaviruses*, the expression of nsp1 inhibits host/reporter gene expression suppression [29,32,33,34,38,39,41,48,49] and plays a role as an antagonist of the IFN response [26,28,50,51,52]. Previous studies have reported nsp1 to be a pathogenic determinant of TGEV, MHV, mouse-adapted SARS-CoV, and SARS-CoV-2 [32,33,34,35,36]. These studies strongly suggest that nsp1 is a target for the development of attenuated live vaccines using gene manipulation of various coronaviruses. Therefore, molecular characterization of BCoV nsp1 may contribute to the development of a novel live vaccine strain against BCoV-mediated diseases, once a reverse genetics system for BCoV is established.

We demonstrated that BCoV nsp1 functions as a shutoff protein, similar to SARS-CoV-2 nsp1 (Figure 2B,D), despite sharing only 24.7% amino acid identity. Shen et al. demonstrated that, although the amino acid identity between SARS-CoV nsp1 and PEDV nsp1 was quite low, the function of host gene expression suppression is conserved [53]. Interestingly, a crystal structure comparison between SARS-CoV nsp1 and PEDV nsp1 showed a high level of structural similarity. Although BCoV nsp1 also has a low level of amino acid identity with SARS-CoV-2 nsp1, the similarity in protein structure may be the reason why BCoV nsp1 and SARS-CoV-2 nsp1 retain shutoff function.

We demonstrated that Lys and Phe at amino acid positions 232 and 233 of BCoV nsp1 are important for inhibiting gene expression in HEK-293 cells and MAC-T cells: BCoV nsp1-KF is a loss-of-function mutant of BCoV nsp1 (Figure 2B,D). We also found that BCoV nsp1-KG and BCoV nsp1-KK suppress plasmid-driven gene expression in MAC-T cells, but not in HEK-293 cells. While the exact reasons behind this observation are not yet fully understood, it might be attributed to species-specific structural differences in host factors (presumably ribosomal proteins) that interact with BCoV nsp1.

Maurina et al. demonstrated that the nsp1 of OC43, a derivative of BCoV, also induces plasmid-based gene expression suppression [54]. We found that BCoV nsp1 shows a relatively high level of amino acid identity with OC43 nsp1 (93.1%). Interestingly, OC43 nsp1 also has Lys and Phe at amino acid positions 232 and 233 (Appendix A), which are important amino acids for host gene expression suppression by BCoV nsp1. This suggests that the amino acids at positions 232 and 233 of OC43 nsp1 are also important for the shutoff function. We believe that this information will be useful for the development of an attenuated live vaccine of OC43 by introducing attenuating mutations at these amino acid positions using reverse genetics systems [55].

SARS-CoV nsp1, MERS-CoV nsp1, and SARS-CoV-2 nsp1 induce the degradation of endogenous host mRNAs in expressed cells [29,38,39]. In contrast, we could not find a reduction in housekeeping gene expression levels in MAC-T cells expressing BCoV nsp1 in the present study (Figure 4). Interestingly, multiple alignment of CoV nsp1s indicates that important amino acids for the mRNA degradation function of nsp1, Arg and Lys at amino acid positions 124 and 125 of SARS-CoV-2 nsp1, are conserved in BCoV nsp1 (Appendix A). This suggests that BCoV nsp1 may retain a host mRNA cleavage/degradation function. Further studies are needed to evaluate the host mRNA degradation function using in vitro assays with purified BCoV nsp1, expressed via *E. coli* or baculovirus systems, similar to studies conducted with SARS-CoV nsp1 and SARS-CoV-2 nsp1 [38,56]. Also, our data do not exclude the possibility that BCoV nsp1-wt may inhibit the de novo synthesis of host RNA in cells.

We found that BCoV nsp1-KF accumulated more prominently in the nucleus, forming distinct dot-like structures, compared with wild-type BCoV nsp1 (Figure 5 and Figure 6). This observation may result in a higher expression of BCoV nsp1-KF than BCoV nsp1. On the other hand, previous studies reported that SARS-CoV-2 nsp1-KH, a biologically inactive mutant similar to BCoV nsp1-KF, localizes to the perinuclear region in expressing cells [36]. However, SARS-CoV-2 nsp1-KH does not form nuclear dots. Thus, the strong nuclear accumulation of BCoV nsp1-KF may suggest a distinct function independent of ribosome binding.

In the present study, we demonstrated that BCoV nsp1 exerts host gene expression suppression by association with ribosomes, similar to SARS-CoV nsp1 and SARS-CoV-2 nsp1 [38,39]. Our data suggest that the suppression of host gene expression is a conserved function of nsp1 among both *Alphacoronaviruses* and *Betacoronaviruses*. Furthermore, we identified the important amino acids of BCoV nsp1 involved in this function. We hope our findings contribute to understanding the pathogenicity of BCoV and lead to the development of novel vaccines against BCoV-mediated diseases in the future.

## Figures and Tables

**Figure 1 viruses-17-00978-f001:**
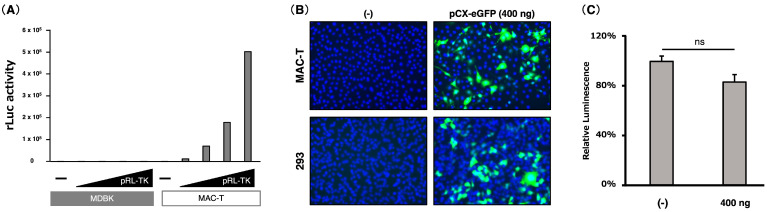
Assessment of the appropriateness of MAC-T cells in transfection experiments. (**A**) MDBK or MAC-T cells were transfected with 0, 50, 100, 200, or 400 ng of pRL-TK (encoding the rLuc gene), and rLuc activities were measured at 24 h post-transfection. (**B**) MAC-T cells were transfected with 400 ng of pCX-eGFP (encoding the eGFP gene), and GFP signals (Green) were observed at 24 h post-transfection. The nucleus was stained with Hoechst 33342 (Blue). (**C**) Cell viability was measured using Cell Titer-GLO at 24 h post-transfection. Error bars show the standard deviations of results from three independent experiments. ns, not significant (*p* > 0.01).

**Figure 2 viruses-17-00978-f002:**
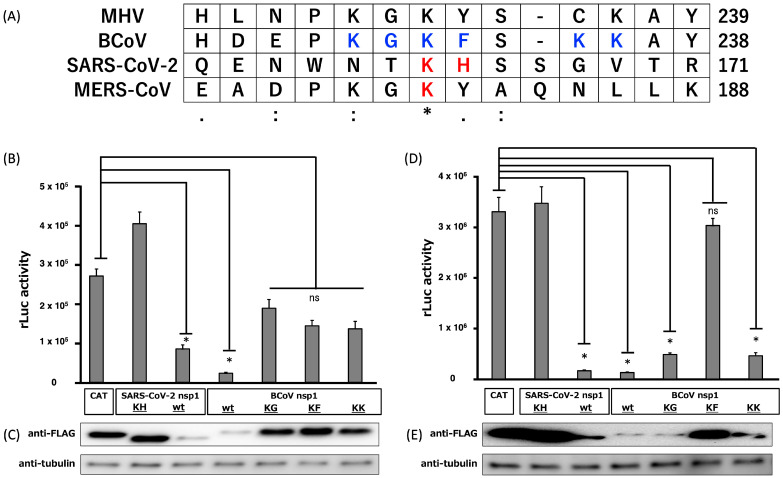
Alignment of the partial amino acid sequences of the nsp1s of MERS-CoV, SARS-CoV-2, MHV, and BCoV, and reporter assay in HEK-293 and MAC-T cells expressing BCoV nsp1 and the mutants. (**A**) Nsp1 sequences of the MERS-CoV strain EMC2012 (accession No.: YP_009047229), SARS-CoV-2 isolate Wuhan-Hu-1 (accession No.: MN908947.3), MHV (accession No.: AF029248.1), and BCoV (accession No.: LC642814.1) are aligned using the Multiple Sequence Comparison by Log-Expectation alignment algorithm. Perfect matches, high amino acid similarities, and low amino acid similarities are represented by asterisks, double dots, and single dots, respectively. A dash “-” indicates a gap in the sequence. The numbers beside the aligned sequences show the positions of amino acid residues. The residues shown in red represent the amino acids that are important for translation inhibition by SARS-CoV-2 nsp1 and MERS-CoV nsp1. The residues shown in blue represent the amino acids in which alanine mutations (KG-to-AA, KF-to-AA, or KK-to-AA mutations) were introduced in the present study. (**B**–**E**) We co-transfected 293 cells (**B**,**C**) or MAC-T cells (**D**,**E**) with pRL-TK (encoding the rLuc gene) and pCAGGS-BCoV nsp1-wt (encoding BCoV nsp1), pCAGGS-BCoV nsp1-KF (encoding BCoV nsp1-KF), pCAGGS-BCoV nsp1-KG (encoding BCoV nsp1-KG), or pCAGGS-BCoV nsp1-KK (encoding BCoV nsp1-KK). As a control, pCAGGS-CAT (encoding the CAT gene), pCAGGS- SARS-CoV-2 nsp1-wt (encoding SARS-CoV-2 nsp1), or pCAGGS-SARS-CoV-2 nsp1-KH mt (encoding biologically inactive SARS-CoV-2 nsp1-mt) was used in place of pCAGGS-BCoV nsp1-wt and -mts. Expressed CAT and nsp1s carried the C-terminal FLAG tag. (**B**,**D**) At 24 h post-transfection, cell lysates were prepared and subjected to a luciferase assay. Error bars show the standard deviations of results from three independent experiments. Asterisks represent significant differences in rLuc activity (*p* < 0.01). ns, not significant. Western blot analysis of the cell extracts was performed to detect plasmid-expressing proteins ((**C**,**E**), top) and tubulin ((**C**,**E**), bottom).

**Figure 3 viruses-17-00978-f003:**
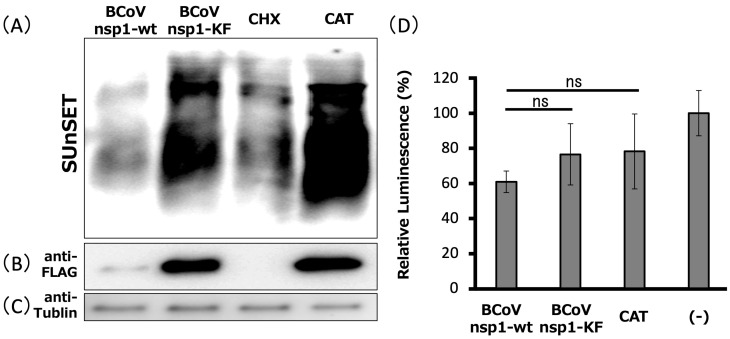
Expression of BCoV nsp1-wt, but not BCoV nsp1-KF, induces inhibition of cellular translation. (**A**) MAC-T cells were transfected with pCAGGS-BCoV nsp1-wt (encoding BCoV nsp1) or pCAGGS-BCoV nsp1-KF (encoding BCoV nsp1-KF). At 24 h post-transfection, cells were incubated for 30 min in media supplemented with 0.5 μg/mL of puromycin. MAC-T cells labeled with puromycin in the presence or absence of cycloheximide were used as the control for translation inhibition. Lysates were resolved using SDS–10% PAGE, followed by Western blot analysis with an anti-puromycin (**A**), anti-FLAG (**B**), or anti-tubulin antibody (**C**), and cell viability assay measured using CellTiter-GLO (**D**). Error bars show the standard deviations of results from three independent experiments. ns, not significant (*p* > 0.01).

**Figure 4 viruses-17-00978-f004:**
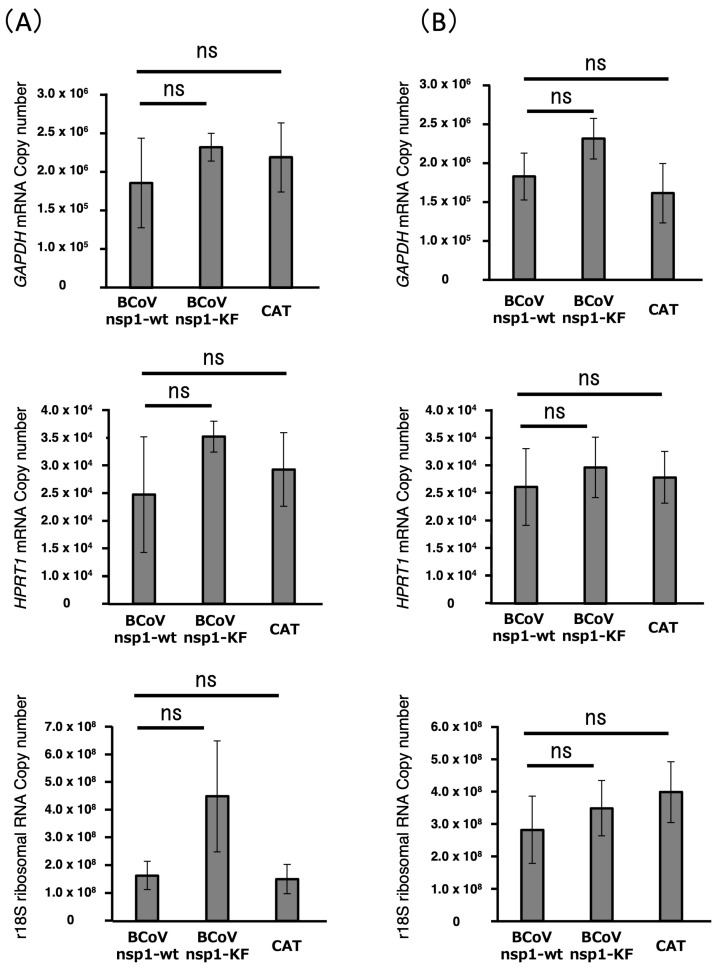
Expression of housekeeping genes in MAC-T cells expressing BCoV nsp1, BCoV nsp1-KF, or CAT. MAC-T cells were transfected with pCAGGS-BCoV nsp1, pCAGGS-BCoV nsp1-KF, or pCAGGS-CAT. At 24 h post-transfection (**A**) and 36 h post-transfection (**B**), copy numbers of *GAPDH* mRNA, *HPRT1* mRNA, and r18S ribosomal RNA were determined by qRT-PCR. Error bars show the standard deviations of results from three independent experiments. ns, not significant (*p* > 0.01).

**Figure 5 viruses-17-00978-f005:**
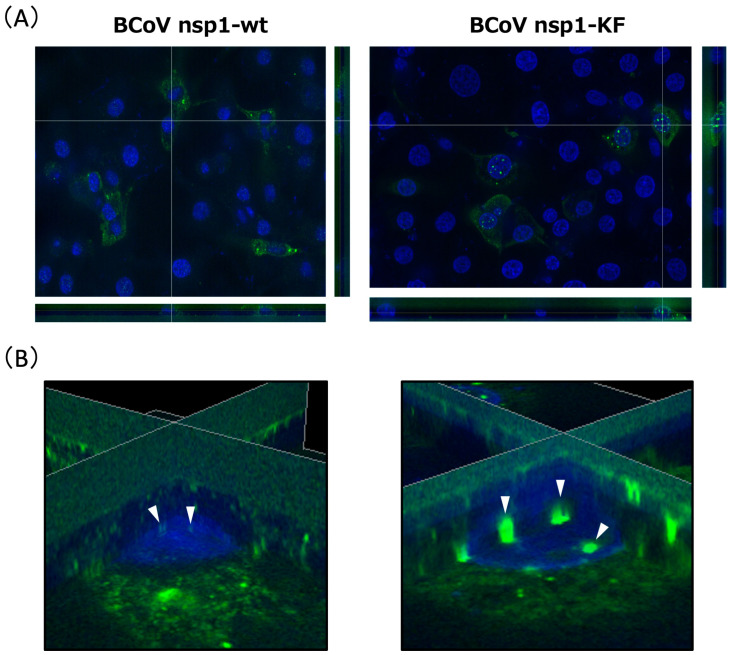
Subcellular localization of BCoV nsp1-wt and BCoV nsp1-KF. MAC-T cells were transfected with pCAGGS-BCoV nsp1-wt (encoding C-terminal FLAG-tagged BCoV nsp1) or pCAGGS-BCoV nsp1-KF (encoding C-terminal FLAG-tagged BCoV nsp1-KF). At 24 h post-transfection, the cells were fixed, permeabilized, and subjected to immunofluorescence analysis with an anti-FLAG antibody. The nuclei were counterstained with Hoechst 33342, and the images were examined using a BZ-X810 fluorescence microscope. For all sets of orthogonal-view images (**A**), the large image shows the X-Y, the bottom image shows the Z-X, and the right image shows the Z-Y view. Zoomed-in view of 3D reconstructed images of BCoV nsp1- or BCoV nsp1-KF-expressing cells (**B**). Arrowheads indicates BCoV nsp1-wt or BCoV nsp1-KF present in the nucleus.

**Figure 6 viruses-17-00978-f006:**
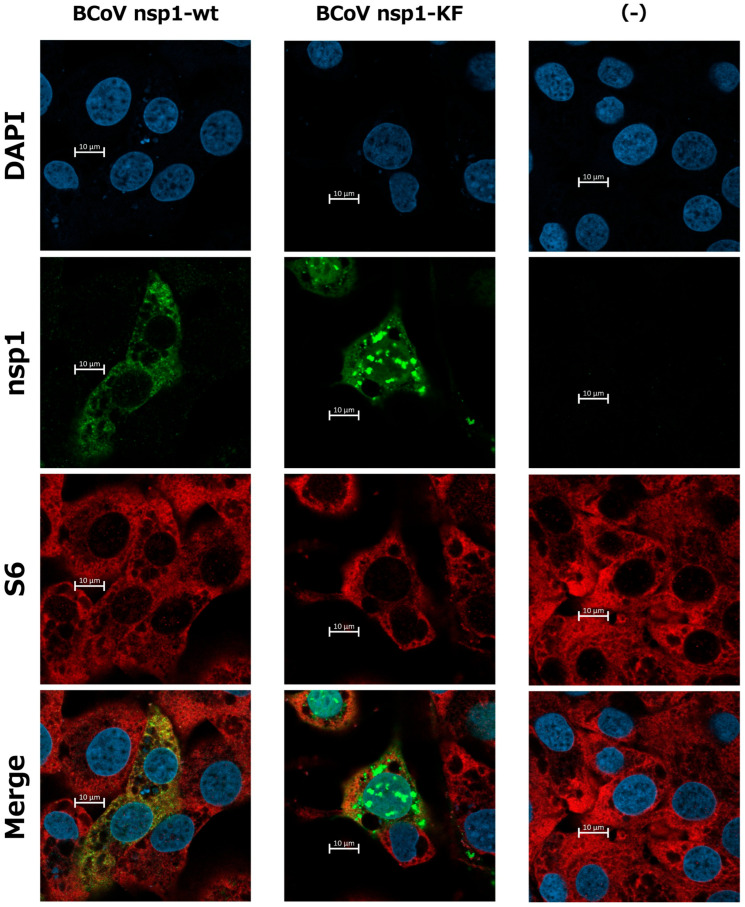
Colocalization of BCoV nsp1-wt, but not BCoV nsp1-KF, with ribosomes. MAC-T cells were mock-transfected or transfected with pCAGGS-BCoV nsp1-wt (encoding C-terminal FLAG-tagged BCoV nsp1) or pCAGGS-BCoV nsp1-KF (encoding C-terminal FLAG-tagged BCoV nsp1-KF). At 24 h post-transfection, the cells were fixed, permeabilized, and subjected to immunofluorescence analysis with an anti-FLAG antibody and anti-S6 ribosomal protein. The nuclei were counterstained with Hoechst 33342, and the images were examined using a Zeiss LSM900 confocal microscope.

**Figure 7 viruses-17-00978-f007:**
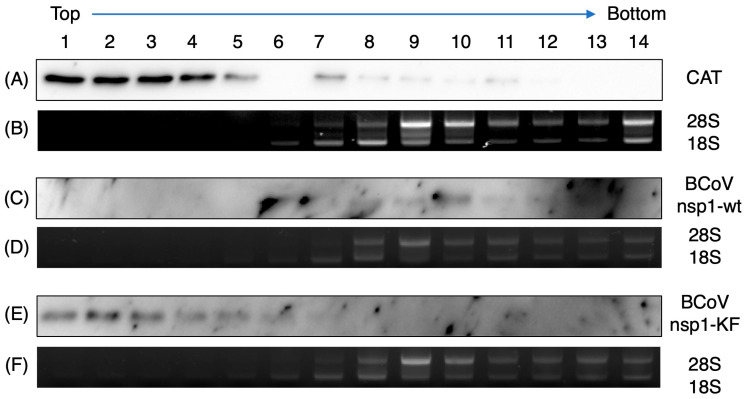
Co-sedimentation of BCoV nsp1-wt, but not BCoV nsp1-KF, with the 40S ribosomal subunit. MAC-T cells were transfected with pCAGGS-CAT (encoding C-terminal FLAG-tagged CAT, panels (**A**,**B**)), BCoV nsp1-wt (encoding C-terminal FLAG-tagged BCoV nsp1, panels (**C**,**D**)), or pCAGGS-BCoV nsp1-KF (encoding C-terminal FLAG-tagged BCoV nsp1-KF, panels (**E**,**F**)). At 24 h post-transfection, the cell extracts were subjected to sucrose gradient centrifugation analysis. The gradient fractions were analyzed by Western blotting with an anti-FLAG antibody to detect the expressed proteins (panels (**A**,**C**,**E**)) and ethidium bromide staining to detect rRNAs (panels (**B**,**D**,**F**)).

## Data Availability

Research data are available from K.N. upon reasonable request.

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
