# Peer review of "Characterization of Gene Expression Suppression by Bovine Coronavirus Non-Structural Protein 1"

_viruses, 2025, doi:10.3390/v17070978_

Round 1

Reviewer 1 Report

Comments and Suggestions for Authors

In this manuscript, the authors investigated the function of Bovine coronavirus Non-Structural Protein 1 (NSP1). More precisely, the study is focused on the host gene expression suppression that is mediated by the viral protein NSP1. Using MAC-T cells, they fist showed that NSP1 is indeed repressing host gene expression and also reporter genes. Then, point mutations at residues 232 and 233 demonstrated that these two residues are critical for efficient translation suppression although housekeeping genes expression remains identical with both wild type and mutated NSP1. A striking difference was the localization of NSP1 proteins, while the Wt NSP1 is found in both the nucleus and the cytoplasm, the mutant proteins accumulates into dot-like structures in the nucleus only. In contrast, the wild-type NSP1, but not the mutant, binds to the ribosome suggesting that binding to the ribosome is a prerequisite for efficient host gene expression suppression. According to the authors, this is the first study dedicated to the host gene repression by the Bovine BCoV NSP1. Although the presented results have potential interest for the virology field, the novelty of the manuscript is very limited. Betacoronavirus NSP1 are known to induce host gene repression in various viruses. More importantly, the manuscript and the presented figures have been prepared in a careless manner. Moreover, the experimental strategy is not appropriate mostly because the NSP1 proteins are tagged at their C-terminal end, which is critical for ribosome binding. Therefore, this reviewer cannot recommend this manuscript for publication at this stage.

Specific points:

- The numbering of residues in Figure 1 is erroneous; the last residues on the right of the alignment are residues #171 and #188 for SARS-CoV-2 and MERS-CoV NSP1 respectively. Moreover, the alignment of MERS-CoV NSP1 to SARS-CoV2 is different to previously published alignments.  

- The plasmids that are used to transfect cells code pour NSP1 protein with a C-terminal FLAG tag. This is problematic because the C-terminal end of NSP1 is folded in two alpha helices that bind tightly in the ribosome mRNA channel. Insertion of a tag at the C-terminal end of NSP1 is detrimental to its function on the host ribosome.

- The loading controls with anti-tubulin antibody are of poor quality in figure 2 and 3. No loading control is presented in Figure 4.

- The statistics in the histograms are puzzling, for instance in figure 5C, the statistics present the results as non-significant, which is very surprising. The same remark applies to Figure 2 and Figure 4.

Typos:

- line 53: replace ‘Embeovirus’ by ‘Embecovirus’

Reviewer 2 Report

Comments and Suggestions for Authors

Remarks to the Authors:

The manuscript by Takehiro Ohkami et al., describes Bovine Coronavirus non-structural protein 1-nsp1 and its role in host shut-off.  Similar to what other labs showed for nsp1 from different CoVs, the authors showed that BCoV nsp1 inhibited host translation and found that the KF residues in positions 232-233 are critical for this inhibition. In addition, they showed the association of nsp1 with ribosomes by confocal microscopy. Although this paper provides important data on BCoV key shut-off factor and makes a contribution to the field, the authors need to address major concerns before this paper is considered for publication in Viruses.

 Major comments:

  1. General: Please consider rearranging the article so that figures 2 and 3 will be combined. A brief report could be a possible solution.
  2. General: In several figures in the paper (such as Figure 4), an empty vector with FLAG tag as control is missing; NT is not a suitable negative control- this must be addressed, and new experiments should be done to cover it, where relevant throughout the paper.
  3. Lines 37-39: …” ribosome binding is required for BCoV nsp1-mediated suppression …” No binding experiments were presented, and only association by IF, so I will suggest rephrasing these sentences to reflect better what was done and be more accurate.
  4. Line 90: This is the first time nsp1-KF and other mutations appear in the paper- it requires more background and more details on these mutations concerning other CoVs- please add this information in the introduction part.
  5. Figure 1: Description of the mutants and the rationale should come first. The figures in the paper should be arranged chronologically, such that the first section in Results discusses the data in Figure 1, and so on. Please rearrange the Results part to address this issue.
  6. Figure 2: Panel B anti-tubulin IB looks bad – please provide a better image.
  7. Lines 105-11: The authors should provide all details on the cloning approach and protocols, including primers used.
  8. Figure 5: In these expression assays, the authors should provide data in M&M on the exact method they used to calculate copy number for the selected targets. In addition, this assay lacks an empty vector control, which is needed. Moreover, measuring the selected transcripts should be done at several time points, not only 24, to get to a clear conclusion. Measuring the level of target gene co-transfected, like GFP is better here. I believe the authors should provide more data here to support their conclusion that BCoV nsp1 does not affect host mRNA as they wrote in lines 278-281.

Minor comments:

  1. Lines 56-57: “The 5′ two-thirds of the genome encodes gene 1…”. Please rephrase these sentences to make it clearer to the reader- Consider using ORF1A/B instead.
  2. Lines 66-68: see previous comment.
  3. Lines 96-99: Please consider adding more information on the cell lines used- source? (gift, ATCC) etc.
  4. Lines 118 and 128: typo? Lipofedtamine?
  5. Line 183: Fig.1 legend, space missing in KF-toAA
  6. Lines 187-200: numbers in the alignments look incorrect – do they reflect the number in each sequence or alignment/consensus? For example, the last residue in BCoV is Y178? Please check and correct if needed.
  7. Lines 97-98: Please provide more details on the cell lines’ growth media – Antibiotics? Co2? Etc
  8. Lines 106-107: “ KG lys to ala and phe to ala..” typo?
  9. Line 97: Please change here and throughout the paper from 293 to HEK-293 to make it clearer to readers.
  10. Line 114: Please add a source of plasmid pRL-TK -Promega?
  11. Figure 2: Please consider changing the y-axis unit (try scientific or log).
  12. Lines 212-213- Does not look like a similar level of tubulin based on the image attached in Figure 2B. Please see a comment regarding this blot in major comments. Please provide a better blot.
  13. Line 198: Figures 2 & 3 before discussing Figure 1 (Lines 214-238) – I suggest keeping it chronological and reordering the Results part.
  14. Lines 232: There is some variation between the different mutants in respect of inhibiting expression, although all are not significant compared to WT, still they vary, so it might be worth discussing or mentioning it.
  15. Lines 234-240: GFP expression in both cell lines looks different, probably based on the transfection efficiency difference between cells- do the authors have empty vector data? In addition, please change the y-axis unit to scientific or log.
  16. Line 244-256: Please refer exact panels in the figures when discussing results.
  17. Figure 4C: Delete or provide a better image or color image.
  18. Line 255: Please consider changing “responsible” to “require”.
  19. Make sure to arrange figures – some figures, such as Fig. 4 and Fig. 5 placed in the middle of a paragraph.
  20. Figure 5: Y-axis units should be changed to log or scientific.
  21. Lines 299-303: No mention of Figure 6B.
  22. Lines 352-360: This part discusses supporting results and should be moved to the first part of the Results section or to M&M section.

Reviewer 3 Report

Comments and Suggestions for Authors

In this manuscript, the authors demonstrated that BCoV nsp1 can inhibit host gene expression and this may be associated with ribosome binding based on the previous results derived from the SARS-CoV and SARS-CoV-2 nsp1. The current results also suggest the suppression of host gene expression is a conserved for both Alphacoronaviruses and Betacoronaviruses. Although the function of nsp1 in gene expression has been extensively studied in Betacoronaviruses, the manuscript is well-structured, the experimental designs and methods are described in details and the results are clearly illustrated. However, some of the conclusions may be overstated based on the current data, and some of the results derived from the figures need to be further addressed for clarity before publication and are as follows.   

  1. Add one sentence to explain the construct of BCoV nsp1-KF because the term appears firstly in the context (in the section of Introduction) of the manuscript (although it has been appeared in Abstract).
  2. Figure (in the figure legend) and Fig. (in the context) need to be consistent throughout the manuscript (all are in the forms of Figure or Fig.). In addition, Figure 1 may be paced after Figure 2 because the Figure 1 is described after Figure 2.
  3. Why the loading control (tubulin) in Figure 2. is not clear (the same control in Figure 3 is good and clear) needs to be addressed. In addition, the authors may explain why the amounts of tubulin is not similar at least by vision (for example, GFP (lane 1) and KK (lane 7) for Figure 2 and lanes 2 and 3 for Figure 3). Are they affected by nsp1?
  4. It is clear that BCoV nsp1 can inhibit host gene expression; however, the authors may further explain why the different results for the BCoV nsp1-mts (KF, for example) are found (in Figures 2 and 3) between in 293 and MAC-T cells.
  5. Kamitani et al. reported that SARS-CoV nsp1 stably binds to the 40S ribosomal subunit and inactivates the translational function of the 40S ribosomal subunit, leading to inhibition of protein synthesis by a series of experiments in that study. In the current study, the authors suggest that BCoV nsp1 colocalized with ribosomes and then conclude that the binding ability of BCoV nsp1 to the ribosome is important to exert the host translation inhibition (lines 310-311). It is suggested that the authors may modify the conclusion because the current data (colocalization of nsp1 with ribosomes) may not be sufficient to support the conclusion (the conclusion may be overstated).
  6. Lines 347-361 may be moved to the section of Results.

Round 2

Reviewer 1 Report

Comments and Suggestions for Authors

The authors addressed my concerns in their revised version of the mansucript.

Author Response

Comment 1: The authors addressed my concerns in their revised version of the manuscript.

Response 1: We deeply appreciate Reviewer #1's helpful comments. Your 

Reviewer 2 Report

Comments and Suggestions for Authors

Remarks to the Authors:

The revised manuscript submitted by Takehiro Ohkami et al. addressed most of my comments, and for that, I would like to thank the authors. However, the authors should address two major comments and a few minor ones before this paper is considered for publication in Viruses.

 Major comment:

  1. Figure 5: Please consider deleting Figure 5. I would recommend performing this analysis on de-novo synthesis of RNA to support the author's conclusion as written in the manuscript.
  2. Figure 3: Figure 2 already described the results of the experiments with the mutants, but Figure 3 is the first time you show the mutants' alignments and describe them- So I believe it will be better to combine these figures such as Figure 3 become figure 2A and then figure 2A become 2B and so on - that will keep things organized chronologically. Please consider changing these figures and the text corresponding to this part in the manuscript.

Minor comments:

  1. Lines 26-27: Please consider rephrasing this sentence in the Abstract: " induces host gene suppression"(to avoid induce vs suppress)- maybe induce host shutoff…
  2. Line 145: typo? Lipofedtamine?
  3. Lines 151-152: The Coomassie image was deleted in the revised version – please delete these sentences in the revised manuscript.

Reviewer 3 Report

Comments and Suggestions for Authors

The authors have responded my concerns  and the reviewer recommends for the acceptance of the manuscript.

Author Response

Comment 1: The authors have responded to my concerns, and the reviewer recommends acceptance of the manuscript.

Response 1: Your comments have enhanced the quality of our paper. We deeply appreciate Reviewer #3's helpful comments.